

# ECMPride: prediction of human extracellular matrix proteins based on the ideal dataset using hybrid features with domain evidence

Binghui Liu[1,*], Ling Leng[2,*], Xuer Sun[3], Yunfang Wang[3], Jie Ma[1] and Yunping Zhu[1,4]

[1] State Key Laboratory of Proteomics, Beijing Proteome Research Center, National Center for Protein Sciences (Beijing), Beijing Institute of Life Omics, Beijing, China
[2] Department of Central Laboratory, Peking Union Medical College Hospital, Peking Union Medical College and Chinese Academy of Medical Sciences, Beijing, China
[3] Tissue Engineering Lab, Institute of Health Service and Transfusion Medicine, Beijing, China
[4] Basic Medical School, Anhui Medical University, Anhui, China
[*] These authors contributed equally to this work.

Corresponding authors
Jie Ma, majie729@163.com
Yunping Zhu,
zhuyunping@gmail.com

## ABSTRACT

Extracellular matrix (ECM) proteins play an essential role in various biological processes in multicellular organisms, and their abnormal regulation can lead to many diseases. For large-scale ECM protein identification, especially through proteomic-based techniques, a theoretical reference database of ECM proteins is required. In this study, based on the experimentally verified ECM datasets and by the integration of protein domain features and a machine learning model, we developed ECMPride, a flexible and scalable tool for predicting ECM proteins. ECMPride achieved excellent performance in predicting ECM proteins, with appropriate balanced accuracy and sensitivity, and the performance of ECMPride was shown to be superior to the previously developed tool. A new theoretical dataset of human ECM components was also established by applying ECMPride to all human entries in the SwissProt database, containing a significant number of putative ECM proteins as well as the abundant biological annotations. This dataset might serve as a valuable reference resource for ECM protein identification.

# INTRODUCTION

The extracellular matrix (ECM) is a vital component of the cellular microenvironment, providing structural and functional support to surrounding cells (*Bonnans, Chou & Werb, 2014*; *Theocharis et al., 2016*). ECM proteins play crucial roles in regulating diverse functions of cells, including differentiation, proliferation, survival, and migration (*Bonnans, Chou & Werb, 2014*; *Hynes, 2009*), and their dysregulation can result in a wide range of diseases (*Bateman, Boot-Handford & Lamandé, 2009*; *Liu et al., 2019*; *Tokhmafshan et al.,*

*2017*; *Walker & Mojares, 2018*). A better understanding of the composition and function of ECM proteins should contribute to useful therapeutic targets for related diseases.

The rapid development of multi-omics research has substantially benefited ECM identification and characterization. However, for large-scale ECM protein identification, especially for proteomics-based techniques, a general reference database of ECM proteins is required. Many strategies have been developed by the researchers to define the set of ECM proteins, including the molecular fishing method (*Cain et al., 2009*), the systematic curation method (*Cromar et al., 2012*), and the domain-based method (*Naba et al., 2016*). Besides, the Richard-Blum lab established the MatrixDB database, which is focused on the interactions established by extracellular proteins and polysaccharides and can provide interaction evidence for putative ECMs validation (*Clerc et al., 2018*). Domain architectures change during evolution (*Apic, Huber & Teichmann, 2003*), and proteins with the same domain architecture are frequently related (*Bornberg-Bauer & Alba, 2013*). By utilizing the domain-based structure of ECM proteins, Naba et al. used an *in silico* approach to define ECM components and, based on this, constructed the Matrisome database in 2012 (*Naba et al., 2012*). The Matrisome has become a general reference database for proteomics-based ECM research in recent years (*Åhrman et al., 2018*; *Gopal et al., 2017*; *Lennon et al., 2014*; *Mayorca-Guiliani et al., 2017*). Further, *Naba et al. (2016)* presented the first draft of the ECM atlas, which was established by integrating publicly available mass spectrometry data from studies explicitly designed to characterize the global composition of ECM proteins. However, when compared with Matrisome, there is relatively low overlap ∼51% (∼73% for Core matrisome and ∼42% for Matrisome-associated) between experimentally identified ECMs and theoretically predicted ones, which likely reflects the poor representation of insoluble matrix tissues in the experimental datasets used for comparison. Additionally, the *in silico* Matrisome was constructed via a semi-empirical and manual-assisted approach, so there are some difficulties for the database in dealing with the problems of constant updating and expansion to other species.

Several attempts have also been made by bioinformatics researchers to predict ECM proteins based on machine learning methods; specifically, a series of tools were developed, including ECMPP (*Jung et al., 2010*), EcmPred (*Kandaswamy et al., 2013*), PECM (*Zhang et al., 2014*), IECMP (*Yang et al., 2015*), ECMP-HybKNN (*Ali & Hayat, 2016*), BAMORF (*Guan, Zhang & Xu, 2017*), and TargetECMP (*Kabir et al., 2018*). Most tools were developed based on a generic pipeline, which uses different machine learning algorithms to build classification models on the extracted features and training datasets and can achieve automated prediction of ECM proteins. The most significant shortcoming of these tools is their lack of a connection with experimental biological features, especially concerning standard dataset construction and classification feature extraction (Article S1). In addition, there are no tools available other than EcmPred.

In summary, the Matrisome database presented by Naba et al. compiles *in silico* and *in vivo* data on ECM proteins, and the existing bioinformatics prediction tools for ECMs are robust in modeling. Thus, in this study, we proposed incorporating these advantages of both approaches and developed ECMPride, a flexible and scalable tool for predicting extracellular matrix proteins. Based on the experimentally verified ECM datasets, while

integrating protein domain features and a machine learning model, ECMPride achieved better performance when compared with EcmPred. We also provide researchers with a comprehensive dataset of all putative human ECMs (named ECMPrideDB) by applying ECMPride to all human protein sequences in the SwissProt database (*The UniProt Consortium, 2017*), and this ECM dataset might serve as a valuable reference resource for future investigations.

## MATERIALS & METHODS

### Datasets

The standard training dataset consists of a positive dataset of ECM proteins and a negative dataset of non-ECM proteins (Table S1). The positive one consists of 521 human proteins whose ECM-related status is supported by Matrisome with further credible evidence (*Naba et al., 2016*) (Table S2). In contrast, the negative one consists of 11,336 human intracellular proteins from the Human Protein Atlas database developed by *Thul et al. (2017)*.

The detailed process of generating positive and negative datasets, as well as the Matrisome categories of the positive dataset, can be found in Article S1.

### Feature extraction

Three main classes and 167 features in total are introduced into ECMPride to represent the characteristics of ECM proteins, including ECM protein-related structural domains (from now on referred to as ECM domains) (*Naba et al., 2012*), physicochemical properties (*Kandaswamy et al., 2013*), and position-specific scoring matrix (PSSM) (*Altschul et al., 1997*) (all features are listed in Table S3).

### ECM domains

We are the first to introduce domain into machine learning algorithms to predict ECM systematically. ECM proteins typically include multiple, independently folded domains whose sequences and arrangements are highly conserved (*Hynes, 2009*). Based on this hallmark, *Naba et al. (2012)* established a list of "inclusion domains" commonly found in ECM proteins and a list of "exclusion domains" whose presence ruled a protein out from being a part of the ECM. These two lists are first merged, and then, domains that are not in the version of InterPro 69.0 (*Mitchell et al., 2018*) or do not exist in any protein of the dataset are excluded. Finally, a list of 63 ECM domains is obtained (Table S3).

The score for $i$-th ECM domain $D_i$ of protein $A$ is represented as follows:

$$X_i = \begin{cases} 0 \ (if \ D_i \in A) \\ 1 \ (if \ D_i \notin A) \end{cases} \quad (i = 1, 2, \ldots, 63)$$

Here, the evidence of whether $D_i$ belongs to $A$ comes from SwissProt (*The UniProt Consortium, 2017*).

Finally, a 63-D feature vector of ECM domains is constructed for every protein sequence.

### Position-Specific Scoring Matrix (PSSM)

For protein evolution, sequences evolve via the substitution, insertion, or deletion of residues (*Chou & Shen, 2007*). After a long time, the accumulation of these changes

slowly eliminates the similarities between the original protein and the final protein; however, some of the critical residues associated with the essential properties of the protein remain stable, which is referred to as evolutionary conservation (*Zhang et al., 2014*). Such conservation usually occurs in sequences with important biological functions (*Zuo et al., 2014*). Therefore, evolutionary information is critical to the prediction of protein structure and function (*Ding et al., 2014*).

PSSM is a matrix that can well reflect the evolution information of a protein. It is generated by running PSI-BLAST (*Altschul et al., 1997*) in the database of SwissProt through three iterations, with 0.001 as an *E*-value cut-off. As shown below, it consists of $20 \times$ L elements, with L representing the length of the protein sequence.

$$P_{PSSM} = \begin{bmatrix} E_{1,1} & E_{1,2} & \cdots & E_{1,j} & \cdots & E_{1,20} \\ E_{2,1} & E_{2,2} & \cdots & E_{2,j} & \cdots & E_{2,20} \\ \vdots & \vdots & \cdots & \vdots & \cdots & \vdots \\ E_{i,1} & E_{i,2} & \cdots & E_{i,j} & \cdots & E_{i,20} \\ \vdots & \vdots & \cdots & \vdots & \cdots & \vdots \\ E_{L,1} & E_{L,2} & \cdots & E_{L,j} & \cdots & E_{L,20} \end{bmatrix}$$

Here, $E_{i,j}$ represents the score of the amino acid mutation in the *i*-th position of the sequence to form the amino acid type *j* during evolution. Then, PSSM is converted into an 80-D vector by standardization and grey model theory (the detailed process of conversion could be found in Article S1) (*Chou, 2001*; *Matsuda et al., 2005*).

## Physicochemical properties

The structure and function of proteins are defined by the physicochemical properties of the 20 amino acids, which have been the subject of a large number of experimental and theoretical studies. The physicochemical properties of the 20 amino acids can be represented by a set of 20 values of an amino acid index (AAIndex) (*Kawashima et al., 2007*). There is now a database exclusively dedicated to storing AAIndex values (UMBC AAindex Database).

Here, we use 24 physicochemical properties selected by *Kandaswamy et al. (2013)* from the UMBC AAindex Database (Table S4). The formula for calculating each physicochemical property of a protein is as follows:

$$PP = \frac{1}{L} \sum_{i=1}^{L} AAIndex_i$$

where $AAIndex_i$ is the AAIndex value of the physicochemical property corresponding to the *i*-th amino acid in the protein sequence, and L is the length of the protein sequence. Finally, a 24-D feature vector of physicochemical properties is established for every protein sequence.

## Feature selection

For feature selection, we first perform feature importance scoring. This involves scoring the importance of all of the extracted features by the Maximum Relevance Minimum

Redundancy (mRMR) algorithm (*Peng, Long & Ding, 2005*) (the detailed process is shown in Article S1). The features are ranked according to the order of the scores from high to low.

Next, we adopt the Incremental Feature Selection (IFS) method to obtain the optimal feature subset based on the ranked feature set. The process begins with an empty feature set and adds features one by one in order of importance from high to low. Each time a feature is added, a new feature subset is generated so that $n$ features will generate $n$ feature subsets (*Lin et al., 2013*). The subset of features with better predictive performance and fewer features would be considered the optimal feature subset (*Yang et al., 2015*).

## Prediction model and Performance evaluation

In this study, the Random forest model has been implemented in ECMPride for prediction. Developed by Breiman, the Random Forest algorithm is an integrated classifier consisting of numerous decision trees. It uses the bootstrap method to extract multiple identical samples from the original sample to generate a training set and then builds a decision tree with each sample in the training set. Finally, the final prediction result of the Random Forest model is obtained by voting on all decision tree prediction results (*Breiman, 2001*). Random Forests have high predictive accuracy, have good tolerance of outliers and noise, and are not prone to over-fitting. They can handle both continuous and discrete variables, making them advantageous and increasingly mature machine learning algorithms. Here we use the randomForest package of R to implement the classification of ECM and non-ECM components (*Liaw & Wiener, 2002*).

Ten-fold cross-validation is used to evaluate the predictive model, and the under-sampling ensemble method is implemented to overcome the imbalance of the training datasets. Meanwhile, we employed the following four parameters to evaluate the performance of the ECM prediction models: sensitivity ($Sn$), specificity ($Sp$), accuracy ($Acc$), and balanced accuracy ($BAcc$). These can be represented by four indicators: true positive ($TP$), false negative ($FN$), true negative ($TN$), and false positive ($FP$). The detailed model training and parameters calculation can be found in Article S1.

# RESULTS

## Construction of ECMPride

We built ECMPride, a command-line based tool that allows users to predict ECM proteins. The tool is available and freely downloaded from the public repository GitHub: https://github.com/Binghui-Liu/ECMPride.git. The overall workflow of ECMPride is shown in Fig. 1. As a high-quality ECM prediction tool, ECMPride has several unique features. First, positive and negative standard datasets (Table S1) are constructed based on reliable experimental and theoretical sources, including the Matrisome, ECM Atlas (*Naba et al., 2016*), the Human Protein Atlas (*Thul et al., 2017*) and Gene Ontology annotation (*The Gene Ontology Consortium, 2016*), as well as a series of ECM proteomic studies (Table S2). Then, three main classes and 167 features in total are introduced into ECMPride to represent the characteristics of ECM proteins. In particular, the ECM domains proposed by Naba et al. are introduced into machine learning algorithms for the first time. In addition,
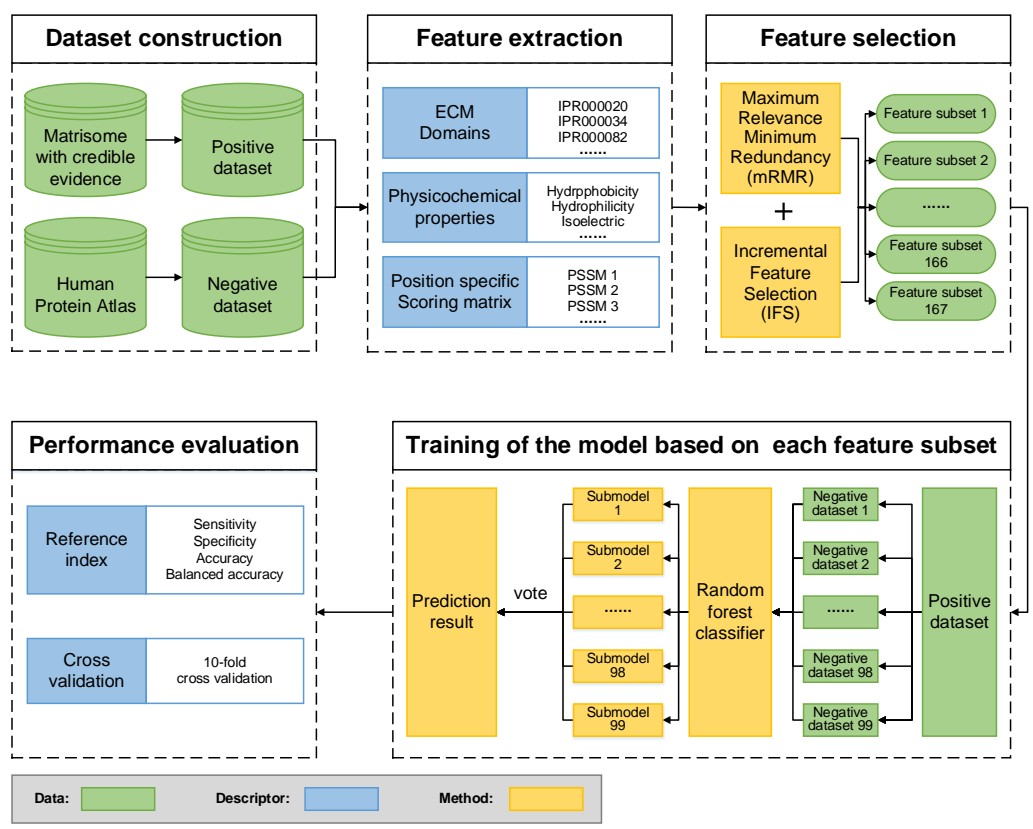

**Figure 1** Flowchart of the ECMPride pipeline.

the mRMR-IFS methods are implemented to reduce feature redundancy. Finally, to handle the classification problem of imbalanced datasets, the under-sampling ensemble method is employed for modeling (Table S5), and balanced accuracy is adopted as the essential criterion to evaluate the performance (Fig. S1). All details about the ECMPride pipeline construction can be found in Materials & Methods and Article S1.

## ECMPride achieves good performance

ECMPride reduces feature redundancy to a certain extent via the feature selection step. All of the 167 features are scored and sorted by mRMR (Table S3), and the IFS method is used to generate 167 feature subsets and further generate 167 corresponding candidate models (Table S6). As shown in Fig. 2, when the top 151 features are selected as the feature subset, the model achieves the highest balanced accuracy of 0.9142, and the corresponding value is 0.9070 when all 167 features are used for prediction. Therefore, feature selection allows us to achieve better prediction with fewer features. In this context, ECMPride is established based on the top 151 features.

A series of tools had been developed by researchers to predict ECM proteins (*Ali & Hayat, 2016*; *Guan, Zhang & Xu, 2017*; *Jung et al., 2010*; *Kabir et al., 2018*; *Kandaswamy et al., 2013*; *Yang et al., 2015*; *Zhang et al., 2014*), so it's necessary to compare ECMPride
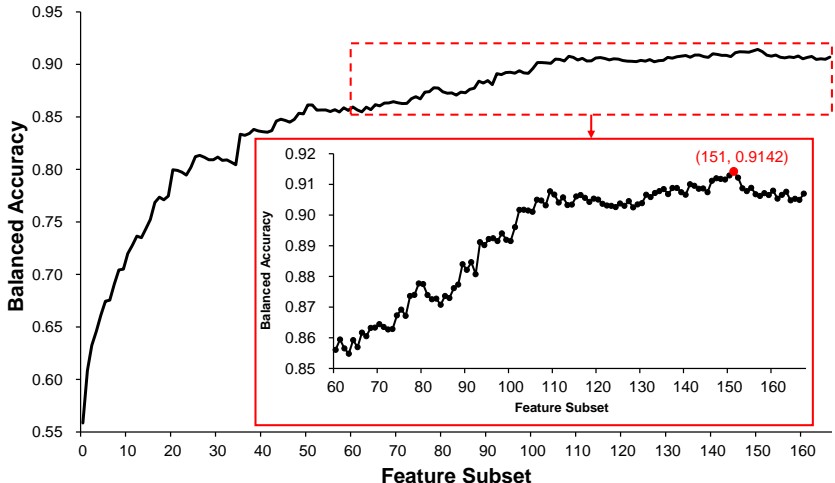

**Figure 2** **The feature selection curve of balanced accuracy for different feature subsets.** The 167 feature subsets were obtained by adding features one by one in order of importance from high to low. On the basis of each feature subset, the model was established with 10-fold cross-validation. The curve represents the relationship between the feature subset and its corresponding model's balanced accuracy.

**Table 1** **Performance comparison of models with different methods and datasets.**

| Method | Dataset | Sensitivity | Specificity | Accuracy | Balanced accuracy |
|--------|---------|-------------|-------------|----------|-------------------|
| ECMPride | D1 | 0.8925 | 0.9360 | 0.9340 | 0.9142 |
|  | D2 | 0.8783 | 0.8623 | 0.8638 | 0.8703 |
| EcmPred | D1 | 0.8462 | 0.9158 | 0.9145 | 0.8810 |
|  | D2 | 0.6500 | 0.7700 | 0.8300 | 0.7100 |

Notes.

D1, Training dataset constructed in ECMPride's model; D2, Training dataset constructed in EcmPred's model.

with these tools. As the datasets used by ECMPride differ from the datasets used for previous tools, it is meaningless to compare their performance directly. Meanwhile, most of the previously released tools are no longer available for a variety of reasons, so it is impossible to compare ECMPride with such tools in an independent dataset. As such, we here attempt to reproduce the previous tools by carefully reviewing the articles about them; only for the tool EcmPred can localization be implemented well (*Kandaswamy et al., 2013*). Therefore, we applied ECMPride's and EcmPred's methods to each other's training dataset and compared their performance (Table 1). Using the same method (ECMPride or EcmPred), the model based on ECMPride's dataset (D1 in Table 1) behaved higher balanced accuracy than that based on EcmPred's dataset (D2 in Table 1), which means that the new training dataset is better than the old one. With the same dataset (D1 or D2 in Table 1), the model based on ECMPride's method behaved higher balanced accuracy than that based on EcmPred's method, which means that the method of ECMPride is better than that of EcmPred. Overall, ECMPride achieved better performance than EcmPred.

## Construction of theoretical reference dataset of human ECM proteins

To obtained a comprehensive collection of theoretical human ECM proteins, we applied ECMPride to all human entries in the SwissProt database (*The UniProt Consortium, 2017*).

The proteins with a probability of being ECM higher than 0.7 are considered to be confidently predicted results. These proteins, together with the positive ECMs, are accepted as putative human ECM proteins and compose the theoretical reference dataset of human ECM proteins (named ECMPrideDB, Table S7). We also collected information on relevant databases to annotate genes in ECMPrideDB (Table S7), including Human Protein Atlas (*Thul et al., 2017*), ExoCarta (*Keerthikumar et al., 2016*) and GO (*The Gene Ontology Consortium, 2016*). Then, we compared ECMPrideDB with Matrisome (*Naba et al., 2016*), as well as two experimental datasets generated from the ECM-related biological samples (Table S8) (*Åhrman et al., 2018*; *Naba et al., 2017*). There are a total of 1,510 putative ECM proteins (1494 genes) in ECMPrideDB, and the official gene symbols were used for comparison with other datasets. Overall, most ECM components in Human Matrisome are included in ECMPrideDB (∼69.62%, Fig. 3A). Specifically, ECMPrideDB covers ∼92.33% of the Core matrisome and ∼61.35% of the Matrisome-associated components in Matrisome (Fig. 3B). Additionally, 779 more novel ECM components are found in ECMPrideDB. For the 21 Core matrisome components uniquely included in Matrisome, 15 of them were also predicted as potential ECMs by ECMPride but with relatively low confidence (probability < 0.7). None of them were annotated with extracellular matrix terms in GO. A similar situation also prevailed in the 291 Matrisome-associated components uniquely included in Matrisome, about half of them (153/291) were predicted as potential ECMs by ECMPride with low confidence (probability <0 .7), and a vast majority of them (275/291) were annotated without extracellular matrix annotations in GO, indicating that there might be insufficient evidence to support these proteins to be real ECMs and more biological or structural information need to be explored.

For both proteomic experimental datasets, most of the identified proteins that overlap with Matrisome are also contained in ECMPrideDB, and considerable numbers of novel ECMs (96 and 127, respectively, Fig. 3C and Fig. 3D) are found in ECMPrideDB.

## Validation of novel ECM components

To further validate the putative ECM proteins predicted by ECMPride, several analyses were implemented. Among the 779 putative ECMs uniquely identified in ECMPrideDB, 283 of them contain at least one of the protein domains proposed by Naba et al. as the specific features for Core Matrisome (*Naba et al., 2016*) (Table S7). It is due to the update of the underlying domain annotation that these newly predicted ECMs emerge. To an extent, it also proves the reliability of putative ECMs predicted by ECMPride. The presentation of experimental interactions with known ECM proteins could be supportive evidence for the new putative ECMs. Thus, the protein-protein interactions of the 779 putative ECMs with all ECMs in Matrisome are retrieved both from MatrixDB (*Clerc et al., 2018*) and STRING (*Szklarczyk et al., 2018*) database. It is found that 619 of 779 putative ECMs can interact with at least one known ECM in Matrisome. Finally, the detailed interactions, as

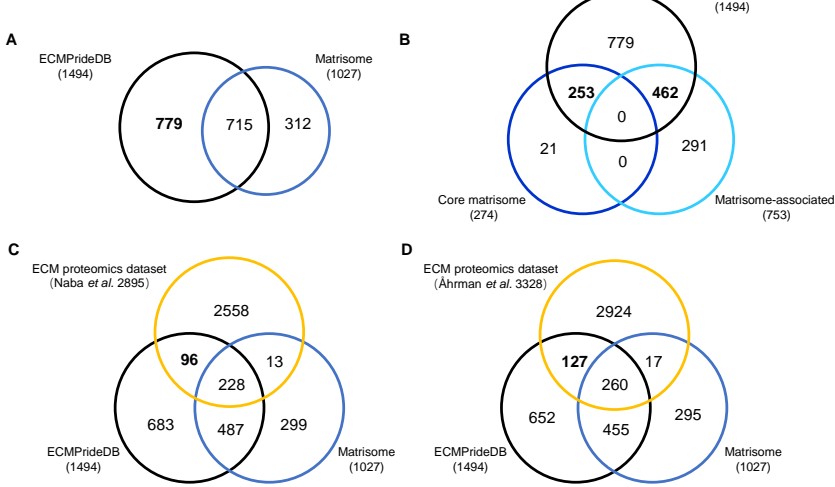

**Figure 3 Comparison of the new ECM proteins with Human Matrisome and other experimental datasets.** The black, blue, and orange circles represent ECMPrideDB, Human Matrisome (dark blue for core Matrisome and light blue for Matrisome-associated), and two proteomics experimental datasets, respectively. (A) Overlap of ECMPrideDB and Matrisome. (B) Overlap of ECMPrideDB, core Matrisome, and Matrisome-associated. (C) Overlap of ECMPrideDB, Matrisome, and ECM proteomics dataset of *Naba et al. (2016)*. (D) Overlap of ECMPrideDB, Matrisome, and ECM proteomics dataset of *Åhrman et al. (2018)*.

well as the hyperlink of the Entrez gene summary, are provided with each putative ECM in ECMPrideDB in Table S7.

Further, we confirmed the expression of several potential novel ECM components in the top list of the ECMPrideDB by immunohistochemistry and immunofluorescence experiments, including stabilin family members STAB1 and STAB2, and the jagged canonical notch ligands JAG1 and JAG2 (Details refer to Article S1). Our results indicate that all four molecules are expressed in the extracellular space of epidermis and dermis (Fig. S2). Interestingly, STAB1, STAB2, and JAG1 are specifically located in the basement membrane of skin tissue, which is the epidermal stem cell niches (Figs. S2A–S2C). And JAG2 is specifically located in the spinous, granular, and stratum corneum layers of the epidermis (Fig. S2D). Moreover, three new predicted ECM components (DLL4, LRP1, and FCGBP) are found expressed in the extracellular space of normal human liver and skin tissues, as well as RH-30 cell lines (Fig. S3). Although the current immunohistochemical and immunofluorescence experiments are not sufficient to verify that these proteins are ECM proteins, the results are nevertheless a useful preliminary validation, and more work remains to be done.

## DISCUSSION

More and more proteomics studies are applied for large-scale ECM protein identification, and the theoretical ECM database was used in these studies to identify ECM proteins and guide the biological analysis and experiments (*Åhrman et al., 2018*; *Gopal et al.,*

*2017*; *Lennon et al., 2014*; *Mayorca-Guiliani et al., 2017*). Thus, the development of ECMs prediction methods and the construct of comprehensive ECM reference datasets are required and will benefit proteomics-based ECM researches.

In this study, we proposed a flexible and scalable tool ECMPride for predicting extracellular matrix proteins by incorporating the advantages of experiment-based features and robust prediction models. There are three classes of features implemented in ECMPride to represent the characteristics of ECM proteins, including ECM protein-related structural domains (63 features), physicochemical properties (24 features), and position-specific scoring matrix (PSSM, 80 features). The physicochemical properties and PSSM have been used in many models and tools for the prediction of protein structure and function for multi-species (*Chen & Li, 2013*; *Du & Yu, 2013*; *Hayat & Khan, 2012*; *Lundegaard et al., 2008*). While for the features of domains, ECM proteins are highly conserved among different species, not only in the sequences of specific domains but also in the arrangements of those domains (*Hynes, 2009*). Utilizing the conserved nature of domains across species, Naba et al. used the same list of domains to construct human and mouse ECM datasets, respectively (*Naba et al., 2012*). At present, we applied ECMPride to predict human ECM proteins, but we think ECMPride can be useful for ECM proteins prediction for other species.

Among all seven ECM prediction tools introduced in this study (ECMPP, EcmPred, PECM, IECMP, ECMP-HybKNN, BAMORF, and TargetECMP), four of them (ECMPP, EcmPred, PECM, and IECMP) were released with web-based applications. Unfortunately, none of these tools are currently available. Therefore, the maintenance and update of software tools are essential for public users. ECMPride is developed as an open-source and easy-to-use tool. To analyze the large datasets efficiently, we also designed a parallel version of ECMPride, which could perform the prediction of proteins with multi-threads mode. All the source codes of ECMPride with single-thread and multi-threads versions are publicly available from GitHub (https://github.com/Binghui-Liu/ECMPride.git). As the experimental validated ECM proteins and annotation database based features would keep updating, we will further improve the sensitivity and specificity of the prediction model and provide the continuously update service of the ECMPride tool. Based on ECMPride, we plan to develop a web-based database for reference ECM proteins for multi-species, which can provide a user-friendly web interface for browsing, searching, and downloading all putative ECM components, as well as the abundant biological annotations.

## CONCLUSIONS

In this study, we developed ECMPride, a flexible and scalable tool for accurate and automatic prediction of ECM proteins. ECMPride can achieve excellent performance in predicting ECM proteins, with a relatively good balanced accuracy and sensitivity. By applying ECMPride to human protein sequences in SwissProt, a new dataset ECMPrideDB of all putative human ECM components was established. This dataset covers most known ECMs in Human Matrisome, and more potential ECM proteins are identified when using this dataset to annotate the experimental proteomics datasets. As ECMPride is developed

based on the machine learning method, the robust of modeling makes it easy to deal with other species' proteins sequences in a similar way, i.e., mouse, rat, and so on. Also, with the accumulation of publicly available ECM proteomics datasets, more experimentally verified ECMs can be added into the standard dataset and further improve the model's prediction performance.

## ACKNOWLEDGEMENTS

We thank Dr. Cheng Chang and Dr. Mansheng Li at the National Center for Protein Sciences (Beijing) for helpful discussion.

### Funding

This work was supported by the National Key Research Program of China (No. 2016YFB0201702, No. 2016YFC0901601, No. 2017YFC0906602, and No. 2017YFA0505002). The funders had no role in study design, data collection and analysis, decision to publish, or preparation of the manuscript.

### Grant Disclosures

The following grant information was disclosed by the authors:
National Key Research Program of China: 2016YFB0201702, 2016YFC0901601, 2017YFC0906602, 2017YFA0505002.

### Competing Interests

The authors declare there are no competing interests.

### Author Contributions

- Binghui Liu conceived and designed the experiments, performed the experiments, analyzed the data, prepared figures and/or tables, authored or reviewed drafts of the paper, and approved the final draft.
- Ling Leng performed the experiments, prepared figures and/or tables, authored or reviewed drafts of the paper, and approved the final draft.
- Xuer Sun performed the experiments, prepared figures and/or tables, and approved the final draft.
- Yunfang Wang conceived and designed the experiments, authored or reviewed drafts of the paper, and approved the final draft.
- Jie Ma conceived and designed the experiments, analyzed the data, prepared figures and/or tables, authored or reviewed drafts of the paper, and approved the final draft.
- Yunping Zhu conceived and designed the experiments, authored or reviewed drafts of the paper, and approved the final draft.

### Data Availability

The code of ECMPride is available at GitHub: https://github.com/Binghui-Liu/ECMPride.git.

## Supplemental Information

Supplemental information for this article can be found online at http://dx.doi.org/10.7717/peerj.9066#supplemental-information.

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
