# Peer review of "ECMPride: prediction of human extracellular matrix proteins based on the ideal dataset using hybrid features with domain evidence"

_PeerJ, doi:10.7717/peerj.9066_

## Round 0.1 · original submission · Major Revisions

Two specialists in the field evaluated your manuscript, and they have several criticisms about this submission. Both reviewers have concerns related to the potential validity of the predictions and requested significant corrections in the paper. In my view, this paper needs a major revision.

Reviewer 1 ·

Basic reporting

The manuscript by Liu et al describes the development of a novel tool for the prediction of extracellular matrix (ECM) proteins to be potentially used as a reference resource for ECM protein identification. Regarding basic features of the manuscript, it is written in a correct, clear way and conforms the standards of courtesy and expression. Moreover, the use of literature references is adequate and, in general terms, it is well structured.

Experimental design

In general, the experimental design is correct. The development of the theoretical tool is well explained and, according to the information provided by authors, is superior to previous tools (Matrisome).

Validity of the findings

To test the validity of the tool to predict ECM proteins, authors applied their algorithm to all human entries in the SwissProt database and compared the results with the most widely used dataset, the human Matrisome. Based on the information provided by the authors, ECMPride detected 706 novel ECM components (not found with Matrisome), while most of the ECM components detected by this tool were also recognized by ECMPride. While the new approach constitutes a remarkable improvement in the detection capacity, there are concerns that should be taken into consideration when throwing conclusions about the performance of ECMPride. To this aim, authors only proposed several candidates and analyzed their expression by immunofluorescence in different tissues, indicating that the tissue distribution is compatible with the localization in the extracellular matrix compartment. This is in fact interesting but, to my opinion, fails short in the attempt to validate their tool.
Theoretically, an ECM component is defined as a protein that is secreted by cells into the extracellular medium and somehow incorporated to the insoluble material that surrounds the cell, together with the cell machinery required to build and remodel this insoluble material. This sophisticated definition can in no way be reduced to a single experiment showing the tissue localization of a particular protein component. In fact, being an ECM component is a dynamic definition that requires a long way validation from the very early identification as a potential candidate to the confirmation by a variety of experimental evidences (analysis of its biosynthesis, secretion, binding to known components, modification, expression as a recombinant protein and many, many more). Software tools such as the one developed by Liu et al are nevertheless tremendously useful in the initial step of identification of potential components, but the manuscript, in its present form, cannot establish this rigorous analysis to confirm that such or such component is in fact ECM. I would definitely remove the IF analysis and concentrate on what I find more useful at using their tool.
For me, the most important point is to perform a more profound comparison with the previous gold-standard approach to identify ECM components, the Matrisome. In other words, I think it is more relevant to discuss why about 700 potential ECM components were detected by ECMPride that were not with Matrisome. Whether they have something in common, hints about their pathophysiological relevance, etc. And the other way around, those components that were detected by Matrisome and not by ECMPride. Why has this occurred? What is it based on? Are they in fact ECM components?

Additional comments

Already discussed in the previous areas above.

Reviewer 2 ·

Basic reporting

In “ECMPride: Prediction of human extracellular matrix proteins based on the ideal dataset using hybrid features with domain evidence” the authors present a method for the in silico prediction of extracellular matrix proteins using a Random Forest classifier trained on features of vetted proteins contained in the Human ECM Atlas. The approach shows a modest improvement over ECMPred and predicts ~600 human proteins not found in MatrisomeDB or other data sets. The authors have validated via immunohistochemistry and immunofluorescence that a small number of putative ECM proteins are indeed present within biologically relevant contexts i.e. the extracellular space. The authors propose that ECMPride will be useful in the prediction of ECM proteins from other genomes and that the generated list of human predictions may be a useful resource to expand our knowledge of ECM function and potential therapeutic targets.

The authors clearly explain their rationale in light of previous work on this dataset and have used acceptable professional English language throughout. Figures are generally of acceptable quality though text and lines in the figures show some rasterization/non-transparency – are these the final figures? It is evident that much more work went into writing technical aspects of the methods than into the biological context and interpretation of the results. Some additional raw data (esp. venn overlaps) should be included as noted below. Please address the following suggestions / questions.

General Concerns:

1. Broadening the scope of the introduction a little, would provide the reader with a better overview of the field and strengthen the rationale. There is tendency to over-rely on just two or three papers which are over-cited. There has been other work in this field to globally define the set of ECM proteins using curation and network methods for example, Cain et al. (pubmed: 19755719) on elastic fibres, Cromar et al. (pubmed: 22275077) on ECM, and various work out of the Richard-Blum lab in France who host MatrixDB (pubmed:20852260). In addition, the seminal paper on protein domain combinations was Apic et al. (pubmed: 14649290) whereas Bornberg-bauer et al. (pubmed: 23562500) covers theory of protein domain evolution.

2. The results/discussion sections are entirely too brief. Much of the text of the paper as a whole is devoted to methods and is largely impenetrable to non-machine learning experts. Matrix biologists will require a logical, lay-explanation of the Random Forest approach, the rationale for choosing this method in light of the available data and commentary assisting their interpretation of the results rather than an impenetrable methods section and simple statement of results.

3. The Discussion section, in its present form, is more of a summary paragraph than a critical consideration of the results and their implications. As such, I suggest that the Results section could be re-titled Results/Discussion and expanded to address some of the concerns raised below. Content currently in discussion could be merged with the concluding paragraph. Otherwise, the discussion section would need to be re-envisioned as a more critical commentary addressing the same concerns.

4. Due to the relatively large number of supplementary files for a paper of this size, I would recommend consolidating the supplementary text into one file of supplementary methods with subheadings.

Detailed comments:

Main text Line 45: Rather than ‘disorder’, I suspect you mean ‘dis-regulation’, or ‘disruption’ via mutation, possibly contributing to ‘mis-assembly’. Disorder is a topic unto itself and quite essential to the normal function of a number of ECM proteins – see e.g. Peysselon et al. (pmid:22009114) on intrinsic disorder in the ECM.

Main text Line 59: The lack of overlap between the matrisome predictions and experimentally identified ECMs was explained in Naba et al. 2016 as largely due to the fact that tissues in which certain matrix proteins were known to be exclusively expressed (e.g. teeth, bones, etc.) were not represented in the experimental data at the time. The difficulty in solubilizing and analyzing such tissues is a reasonable limitation the authors faced. Rather than making a sweeping criticism about the lack of overlap (which was still >60% for the curated set and 50% overall for the predicted set) you should qualify your remarks by stating the facts (i.e. a low overlap ~ 50-60% which likely reflects the poor representation of insoluble matrix tissues in the experimental set used for comparison). Still, this reinforces the point that there is an opportunity for in silico methods to circumvent these significant experimental limitations!

Main text Line 65 – 74: The discussion of pre-existing methods is appreciated, especially the more detailed discussion in Article S1. Could the authors include a comparison of the results of these tools in terms of the sensitivity and specificity or, total numbers of ECM proteins predicted at a particular confidence level? Without this, it is difficult to appreciate the relative contribution of these advances in methodology or, the assertion that, (Line 73) “their lack of a connection with experimental biological features” is a significant drawback.

Main text Line 297-298: Pertaining to Figure 3, please refer to your own subheadings when referring to your observations. For example, in Figure 3a (second venn), your dataset does not cover nearly a quarter of the core matrisome which, are a highly curated set. This deserves comment. As a suggestion, it may be more succinct to label your dataset simply as “ECMPride” within these figures and, more correct to refer to ECM proteins you have predicted as “putative ECM proteins” since the predictions are unverified. Also in Figure 3a (first venn), since the Matrisome comprises both curated and predicted data it would be nice to see how much of ECMPride overlaps these two categories within the Matrisome. In supplementary material, please provide the raw data for these figures, namely the list of proteins in each section of each venn.

S2 Line 6-8: It is difficult to tell whether the main reference here is intended to be Ulen et al., Thul et al. or both?

Experimental design

Given the limitations of the data, the authors are to be applauded on their ambition and creativity in seeing relevant dimensions on which to base their classifier. Having said this, the following concerns need to be addressed:

General Concerns:

1. It is not entirely clear what classes of genes are represented in the positive ECM set. The authors state that they used genes derived from the Human ECM Atlas corresponding to 478 proteins defined by Swiss-Prot identifiers. It would be helpful if the authors provided a characterization of the composition of the data similar to Fig 3 in Nada et al. 2016 but showing the number of genes in the positive data set comprising each matrisome category.

2. The purpose of using the Atlas was to base the positive set on experimentally supported ECM assignments. However, there are two notable problems with this.
a) First, the atlas is partly made up of ECM data derived from tumor tissues, reflecting abnormal proteins and ECM compositions. Did the authors include or exclude proteins derived on the basis of only tumor data?
b) Second, we know that the Atlas does not include a significant number of bona fide ECM proteins that are known to be exclusively expressed in tissues like teeth, bones etc. that are resistant to solubilisation (a requirement for MS analysis). This means that the positive data set is (unavoidably) biased. This should be acknowledged. It also means that any bona fide ECM proteins that are known not to be present in the Atlas need to be either added to the positive data set or, removed from consideration as novel predictions when evaluating ECMPride.

3) Given the large structural/functional diversity of proteins comprising the ECM, I am surprised that a PSSM would provide much useful information. Even among collagens there is enormous sequence/structure/function diversity. Can the authors clearly state how much weight was given to the evolutionary sequence conservation vs. other factors? It is difficult to see this having a large influence given that other factors like domain composition, presence of a signal peptide or transmembrane domain (all of which can be reliably predicted) are themselves quite diagnostic. Have the authors considered breaking their ECM positive set into different categories, similar to Nada et al. to see whether predictions based on an ensemble of category-specific classifiers provides increased predictive power?

Detailed comments:

Main text Line 253: Re: domains being introduced for the first time. Are the domains not implicitly represented in the PSSM data already and therefore included in previous methods? Arguably, they represent the least noisy and most diagnostic information in your sequence based data. Could this be the reason that their inclusion adds little more improvement over the previous PSSM based methods? i.e. Is it consistent with your results to suggest that all you have effectively done is boost the signal to noise ratio by spiking in a repeat of the most diagnostic, most conserved sequences?

Table S6: Among ECMPride predictions that are not found in MatrisomeDB, there appears to be a very high enrichment for membrane proteins. Perhaps this is due to the exclusion of membrane-associated genes from the negative training set. Would it not be better to include membrane proteins in the negative set and screen them against Matrisome for identified ECM genes as you did for intracellular genes? Please discuss your choice here.

Validity of the findings

The authors have applied a rigorous methodology and confronted important and challenging issues such as the imbalance in the size of the positive and negative training sets as well as construction and optimization of features used for prediction. As with any predictive method, the quality of the predictions is necessarily limited by imperfections in the training data and, our evaluation of them, by the present state of annotations. These limitations are not explicitly acknowledged by the authors, leading to some over-reach in their claims. Nevertheless, the work herein remains a meritorious advancement on previous methods. Please address the following comments / concerns:

General Concerns:

1. The experimental immunohistochemistry and immunofluorescence results provide validation that a very select number of ECMPride predicted proteins are expressed as extracellular proteins. They do not confirm that they are ECM proteins and here, the authors are appropriately careful not to over-reach in drawing their conclusions. As a suggestion, the presentation of any experimental interaction data for these proteins might help bolster support for the functional associations that are suggested here, particularly if they establish a physical interaction with known ECM proteins. For this, I would suggest looking at MatrixDB as a starting point.

2. Due to the difficulty in assessing the potential validity of the predictions at a glance, I recommend providing additional links in Table S6 to aggregated functional information. One solution would be to provide a hyperlink to the corresponding entrez gene summary for each gene.

3. The suggestion that this method could be useful if applied to other species should include the qualification that its performance would depend on the degree to which similar features are conserved in that group.

Detailed comments:

Main text Line 81-82: The conclusion that “ECMPride achieved relatively good performance in ECM protein prediction” is a generalization that unfortunately, is weakened by the lack of comparisons with previous tools. At most the authors can claim a modest improvement in performance over ECMPred and refer to the specific measures of sensitivity and specificity that they achieved. On the other hand, the authors may wish to emphasize other aspects of their software tool such as availability, ease of use, the robustness of the underlying training data (albeit with some biases)– as improvements over previously available tools.

Main text Line 257-258: Figure S1 contains interesting inflection points. Can the authors provide any insight into the nature of the feature sets (1-25) vs. (30-100) vs. (101 – 146) representing the corresponding inflections? In addition, other than mention of minimal requirements listed in the user guide, I did not see any performance data associated with running the algorithm. It might be advisable to give some indication of the performance (i.e. run times) relative to the number of feature sets and to compare this with the performance of ECMPred. Are there trade-offs here that would become relevant over larger datasets, say, all sequenced genomes?

Main text Line 278-279: Per “ECMPred is taken as a representative of the previous tools”. Is it the case that ECMPred was the best of the preceding tools? Or, is it simply that it is the only one available and you propose it is a suitable stand in? I do not see how ECMPred can represent any other tool than itself, nor does it necessarily need to. Simply state why the other tools are unsuitable (or unavailable) and scope your conclusions to the comparison you were able to perform.

Main text Line 299-300: There is no data presented to support the observation that 263 ECMPride predicted proteins contain at least one ECM specific domain. Nevertheless, does this really represent an improvement in prediction due to ECMPride? Or, could it perhaps reflect a subsequent change in the underlying domain annotations since the last Matrisome update?

---

## Round 0.2 · Minor Revisions

One of the reviewers asked for a minor revision as indicated in his/her evaluation.

Reviewer 1 ·

Basic reporting

These aspects are correct.

Experimental design

Changes included in the revised version have substantially improved the manuscript.

Validity of the findings

Raised concerns are adequately answered and corresponding changes included in the revised version.

Additional comments

Raised concerns are adequately answered and corresponding changes included in the revised version.

Reviewer 2 ·

Basic reporting

The authors clearly explain their rationale in light of previous work on this data set and have used acceptable professional English language throughout.

Concerns:

1. Line 330 contains a misspelling of “ECMPrirde”
2. Lines 238-242 is a little vague. Rather than saying “better” I recommend walking the reader through the specific results shown in the corresponding table.

Experimental design

No additional concerns.

Validity of the findings

The authors have adequately responded to the concerns raised in the initial review.

Additional Concerns:

1. Line 266 – Perhaps the limitation is with the GO annotation itself ?
2. Lines 268-270 – You do not discuss the (very) large non-overlap with the additional data sets in Fig 3b,c (Naba and Ahrman) which are striking! Are they merely false positives?

Additional comments

I congratulate the authors on their thoughtful and resourceful responses to the initial review. Overall, this is a much more sound and informative paper in its present form.

---

## Round 0.3 · accepted · Accept

The authors modified the manuscript accordingly with the suggestions made by the reviewers. In my view, the manuscript improved during the revision process and can be accepted as it is.